# Application of Artificial Intelligence at All Stages of Bone Tissue Engineering

**DOI:** 10.3390/biomedicines12010076

**Published:** 2023-12-28

**Authors:** Ekaterina Kolomenskaya, Vera Butova, Artem Poltavskiy, Alexander Soldatov, Maria Butakova

**Affiliations:** 1The Smart Materials Research Institute, Southern Federal University, 178/24 Sladkova, 344090 Rostov-on-Don, Russia; vbutova@sfedu.ru (V.B.); poltavsky@sfedu.ru (A.P.); soldatov@sfedu.ru (A.S.); mbutakova@sfedu.ru (M.B.); 2Institute of General and Inorganic Chemistry, Bulgarian Academy of Sciences, 1113 Sofia, Bulgaria

**Keywords:** scaffolds, artificial intelligence, bone implants, machine learning, screening, biomedical materials

## Abstract

The development of artificial intelligence (AI) has revolutionized medical care in recent years and plays a vital role in a number of areas, such as diagnostics and forecasting. In this review, we discuss the most promising areas of AI application to the field of bone tissue engineering and prosthetics, which can drastically benefit from AI-assisted optimization and patient personalization of implants and scaffolds in ways ranging from visualization and real-time monitoring to the implantation cases prediction, thereby leveraging the compromise between specific architecture decisions, material choice, and synthesis procedure. With the emphasized crucial role of accuracy and robustness of developed AI algorithms, especially in bone tissue engineering, it was shown that rigorous validation and testing, demanding large datasets and extensive clinical trials, are essential, and we discuss how through developing multidisciplinary cooperation among biology, chemistry with materials science, and AI, these challenges can be addressed.

## 1. Introduction

Ever-growing social development and the population aging pose a great demand for advanced novel approaches for a patient’s treatment, including bone tissue restoration as one of the primary clinical and socioeconomic needs; currently, bone defects and functional disorders represent one of the global health problems [1]. Over the last 30 years, medicine has been actively developing, thereby leveraging modern methods of treating bone injuries, which are largely focused on replacing lost bone with allogeneic or autogenous bone grafts, which are limited in many aspects [2].

Considering autogenic bone grafting, the amount of donor tissue and frequent complications at the donor site are the limiting factors. Regarding the case of allogeneic bone grafts, cell-mediated immune reactions and possible pathogen transfer constitute a significant problem [3].

The small volume of autogenic bone material, the long operating time, the risk of high blood loss, the morbidity of donor sites, and the large invasiveness of the method complicate the treatment of extensive bone defects with this approach [4]. On the other hand, low osteogenicity, possible immunological reactions, and low mechanical strength challenge the development of a synthetic material for these tasks. Nevertheless, tailoring the porosity, biodegradation rate, bioactivity, osteoinductivity, osteoconductivity, and mechanical properties of composites can address a range of problems in orthopedics.

The selection of the material for a bone implant can be approached in various ways. Each material has a set of advantages and disadvantages, from which the optimal option should be chosen for each specific task [5].

Materials for bone tissue engineering can be divided into biotolerant, bioinert, biodegradable, and bioactive according to their biocompatibility. Biotolerant materials demonstrate acceptable biocompatibility with bone tissue but lack osteoconductive properties. Although protein adhesion is observable on their surfaces, forming a secure contact with bone tissue through osseointegration is not possible [6]. Such materials are kept separated by the development of fibrous tissue induced by the release of ions and chemical compounds from the implant. This category includes nearly all synthetic polymers and most metals [7,8,9,10]. Bioinert materials are stable and do not interact with body fluids. Similar to biotolerant materials, these implants are separated from the tissues. However, under specific conditions, bioinert materials can establish a direct functional or structural connection with bone tissue [11,12]. Biodegradable materials dissolve in contact with body fluids. The decomposition products of their materials are nontoxic and can be secreted through the kidneys. Bioactive materials can form bonds with bone tissues directly, thereby creating an environment compatible with osteogenesis [13,14,15,16,17,18,19]. These types of materials can be subdivided into osteoconductive and osteoinductive. The former can stimulate the development a of new bone.

Materials for bone tissue engineering can be divided into inorganic, organic, and composite materials according to their composition. Bioceramics based on aluminum oxide and zirconium oxide are characterized by excellent biocompatibility, osteoconductive properties, and corrosion resistance [5,17,19,20]. These materials have been successfully used in orthopedics, especially for total hip and knee arthroplasty, due to their chemical bioinertness, as well as high strength, hardness, and resistance to cracking and corrosion [5]. Bioactive glasses are another important class of bioceramics based on silicates. The 45S5 (bioglass) type contains 45 wt% SiO_2_, 24.5 wt% CaO, 24.5 wt% Na_2_O, and 6.0 wt% P_2_O_5_. The 45S5 type provides a higher bone regeneration rate compared to bioactive ceramics based on Hap. After the implantation of bioactive glasses, they form a layer of amorphous calcium phosphate or HAp on the implant surface [21]. This leads to an increased rate of tissue binding. Moreover, the release of Si, Ca, P, and Na ions from silicate glasses upon dissolution stimulates osteogenesis, neovascularization/ angiogenesis, and enzymatic activity. This has drawn significant attention, as they can be used to create functional materials. For example, Sr-doped bioactive glass facilitates early in vivo angiogenesis by altering the phenotype of inflammatory macrophages, while Ag-doped bioglass showed a clear antimicrobial effect against both Gram-negative and Gram-positive bacteria [22]. Additionally mesoporous bioactive glasses can be loaded with various drugs or biomolecules [23]. Natural biopolymers used as materials for the restoration of damaged bone tissues include collagen, gelatin, chitosan, hyaluronic acid, silk fibers, and alginate [5,23,24]. All of them are characterized by biocompatibility and biodegradability. These materials are typically hydrophilic and form hydrogels with high water content. Natural biopolymers are subject to preimplantation mineralization, which potentially makes them osteoinductive. However, obtaining natural biopolymers involves using natural sources, thus raising concerns about the variability of composition that affects their commercialization. Moreover, there are less options to modify these materials to compensate for the low mechanical strength of natural polymers [23].

A promising alternative to natural polymers is biodegradable synthetic polymers [25]. They are characterized by very high strength and stiffness, which are necessary for the regeneration of bone tissues. The key factors of their degradation rate are regulated by molecular weight, chemical composition, and crystallinity. It is essential to select an optimal composition to avoid a bulk erosion process, thereby leading to premature breakdown of scaffolds and even the sudden release of acidic degradation products that can cause a strong inflammatory reaction [5]. In comparison to natural polymers, synthetic polymers offer more opportunities for chemical modifications and molecular changes. For example, the properties of hydrophilicity and hydrophobicity in synthetic polymers can be regulated, thereby influencing their interaction with the physiological environment. The absence of conjugated motifs that can control cell behavior in hydrophobic synthetic polymers leads to the predominance of osteoconductive rather than osteoinductive characteristics. However, hydrophobicity reduces their immunogenicity [23].

Another promising direction is the production of composite materials containing both organic and inorganic components. Combining different components in a composite allows for the creation of a material with mechanical and biological properties that have significantly better properties than those of individual components [5]. In recent work, magnetic mesoporous bioglass was combined with organic polymer poly(3-hydroxybutyrate)-chitosan [26]. The authors produced ultrathin nanofibers with enhanced mechanical properties. Another report showed that the incorporation of barium-doped baghdadite bioceramics into the polymer scaffold (Poly(3-hydroxybutyrate-co-3-hydroxyvalerate) and poly(ε-caprolactone)) resulted in increased bioactivity and higher cell proliferation [27].

The vertebrate bone matrix is an intercellular substance of bone tissue with a high concentration of calcium salts. The main mineral component of the vertebrate bone matrix is the mineral hydroxyapatite (Ca_10_(PO_4_)_6_(OH)_2_) [28,29]. Therefore, HAp-based ceramics are biocompatible, bioactive, and biodegradable in the body of the patient. Implants based on such ceramics are an excellent alternative to autografts and allografts [30].

It has been scientifically proven that HAp can promote the ingrowth of new bone through the mechanism of osteoconduction without causing any local or systemic toxicity, inflammation, or reaction to a foreign body [31,32,33,34].

HAp has been widely used in rapidly developing bone tissue engineering [35,36], thereby being a prominent alternative in one of the most promising areas of targeted drug delivery [37]. The synthesis and application of HAp specifically for biomedical applications [38,39] requires scientists to solve many problems. The main problem is the choice of parameters for the synthesis of materials.

From a drug delivery point of view, developing and improving ceramics for use in bone implant applications have become prominent [40]. The main directions were the increase in the surface area of the material, as well as the production of porous scaffolds. Advanced techniques were used to enhance control over the material architecture. Among them, 3D printing and electrospinning offer simple ways to construct customized implants with complex structures [41,42].

Bone scaffolds can provide mechanical properties akin to natural bone. During treatment, the frame material is replaced by the patient’s bone tissue without the loss of mechanical properties. The use of growth factors in porous and bioresorbable scaffolds stimulates cell attachment proliferation and differentiation, and it accelerates tissue regeneration [43]. Furthermore, the ability of the framework to deliver biologically active molecules has an effect on improving tissue regeneration [44].

Various requirements and characteristics are imposed on the scaffold material; it must have good bioactivity, biodegradability, biocompatibility, and a suitable porous structure that ensures cell proliferation, vascular germination, suitable surface morphology, and other physicochemical properties [45,46].

To obtain an ideal scaffold, researchers previously had to use the trial-and-error method, as shown on Figure 1, thus spending a lot of time and reagents. Today, artificial intelligence (AI), including machine learning (ML), can provide alternative routes for materials development and optimization [47].

The integration of modern computational methods for modeling the optimal morphological and bioactive properties of scaffolds using AI plays a significant role in the development of this research topic. The roadmap of AI application in medical chemistry is shown on Figure 2. The most common and proven in-practice technologies are numerical modeling, deep neural networks, knowledge-based recommendation systems, computer vision, machine learning algorithms, solving table-based regression, and classification tasks.

In the last decade, the volume of biomedical data has increased dramatically, and the development of medical technologies has jumped into the opportunities provided by the computer era [48]. To search for information qualitatively and effectively, it is necessary to choose an appropriate approach to process biomedical data. Deep learning (DL) is a well-proven, effective method that surpasses traditional machine learning (ML) models in a number of areas such as computer vision [49]. Today, tools based on artificial neural networks (ANNs) have shown a high potential to predict the behavior and properties of alloys [50]. Using numerical modeling methods, it is possible to investigate the effect of bone ingrowth on the mechanical properties of titanium alloy frames [49]. The mechanical properties of the scaffold material, considering its microarchitecture and matrix material properties, can be predicted using the finite element method [50].

These approaches can significantly reduce resource and time costs, as well as increase the efficiency and speed of the production of scaffolds with unique properties for the end user. The use of artificial intelligence in the synthesis and production of skeletons makes it possible to simplify screening and diagnostic procedures significantly [51], optimize experimental parameters needed to obtain the desired structural and bioactive properties of materials [52], develop reliable models of the regeneration of living tissues [53], and study the survival of the implant after installation [54].

In this paper, we discuss the most promising areas of the application of artificial intelligence. Because biomedical data are difficult to access and process, neural networks can assist scientists in analyzing and processing large datasets. Artificial intelligence contributes significantly to the healthcare sector by identifying diseases and determining the best treatment methods for patients.

In this review, we address the following questions:At what stages of tissue engineering is artificial intelligence used?What AI tools do scientists and medical professionals use?How does artificial intelligence help reduce the material’s development time?Most importantly, how does AI improve a patient’s quality of life?

## 2. Prediction of Implantation Cases

Despite the improvement in implant manufacturing technologies and the introduction of new diagnostic protocols, there is a need to develop new methods for the preliminary assessment of the condition to predict the successful survival of a single implant. The intensive development of artificial intelligence (AI) technologies and the increased amount of digital information available for analysis make it relevant to develop systems based on neural networks for auxiliary diagnostics and forecasting. These methods are currently being actively used in various healthcare fields [55].

The method of thematic modeling is used to structure digital information. Several methods for topic modeling have been proposed in recent years, such as latent Dirichlet analysis [56] or Top2vec [57], which is based on Word2vec [58]. All of these methods use BERT embeddings; BERT stands for bidirectional encoder representations from transformers and is a machine learning framework that is based on transformers, which are a deep learning architecture [59,60].

AI has revolutionized medical care in the last decade and plays a vital role in a number of areas, such as diagnostics and forecasting. Several methods have been used in recent years in medical practice to prevent fractures [61]. Previously, osteoporosis was already determined surgically by the T score from a bone mineral density (BMD) test [62]. BMD measured using dual-energy X-ray absorptiometry (DXA) reliably predicts serious fractures [63]. The prognosis improves if combined with additional clinical risk factors. Several risk prediction tools such as the Fracture Risk Assessment Tool (FRAX) [64], the Garvan Fracture Risk Calculator [65], and the QFracture score [66] have been developed.

Earlier AI models used in medicine were based on logic and symbolic methods [67]. They lacked the accuracy and predictive capabilities of modern algorithmic models related to ML and DL. ML is a type of artificial intelligence that allows machines to learn from data without explicit programming. Its algorithms, such as Bayesian networks, ensemble methods, and gradient boosting algorithms, are actively being used to contribute to the health sector by discovering diseases and determining the best treatments for patients, such as, for example, for dental implant necessity prediction [68].

DL is a subset of ML that adds up more layers in artificial neural networks (ANNs), thereby increasing the model capacity to capture patterns in the data and solve more complex tasks. ANNs represent a class of computational models inspired by the human brain’s neural architecture and comprising interconnected nodes or artificial neurons organized into layers. Each neuron processes information by summing weighted inputs, applying an activation function, and transmitting the result to subsequent neurons. The network’s connections, governed by adjustable weights, allow it to adapt and learn patterns from data. ANNs have been applied in various fields due to their capacity to model complex relationships and patterns in data. ANN models help implantologists pay attention to minor factors that affect the quality of the installation, predict the future survival of the implant, and reduce the percentage of complications in all stages of treatment [69]. Implementing a neural network for predicting the survival rate of single implants with a test accuracy of 94.48% contained an ANN model trained on more than 1600 patients’ data using the ReLU and the softmax activation functions for probabilistic distribution. The model used 55 one hot-encoded statistical factors of patients’ data to predict the possibility for two recognition classes of implants: “survival” and “rejection”.

## 3. Selection of Candidate’s Material

Choosing the material for implantation is a nontrivial task. Obtaining a candidate drug often requires the simultaneous improvement of various properties of the compound. The pharmacophore modeling concept is simple and gives precise results in many ways. A pharmacophore captures the spatial locations of the features and rigidly models the interaction between the ligand and its binding site in a specific binding situation. The result is a three-dimensional (3D) spatial arrangement of chemical elements, which is derived using algorithms that consider rules derived from knowledge of chemistry [59]. The technology to obtain three-dimensional pharmacophores is an essential method of drug detection [70]. To preserve the good properties of the chemical series and remove the bad ones, replacing the parts of the molecule responsible for the undesirable properties is necessary. The terms rescaffolding and scaffold hopping are used to describe this replacement of the central structure of the nucleus of a molecule with another chemical motif [71,72,73,74]. Scaffolding can also be achieved using virtual screening. An abrupt change in the structure with the help of virtual screening can lead to the production of compounds with increased biological activity. Often, the structure replacement is accompanied by a decrease in activity.

The choice of material is also influenced by the additional method of creating implants. Currently, a method combining 3D printing and robotics has been used to provide the mass personalization of orthopedic implants [75]. For example, bioprinting was applied to produce patient-specific shaped heart valves [76]. Various titanium alloys are considered to be among the leading materials for bone tissue engineering [77]. Another promising class of materials are various polymers of natural or synthetic origin [44]. Polymers are used to make porous frameworks with the function of releasing drugs. As a part of composites, polymers can improve the mechanical properties of materials. Electrospinning is another method used to develop regenerative skeletons for osteogenesis that mimic the structures and components of natural bone tissue by selecting appropriate custom synthetic or biomimetic natural materials, including metals and composites, ceramics, and polymers. Electrospinning is a simple method that can produce cell-attached scaffolds with large surface areas, a high distance between fibers for cell gas exchange, infiltration, and nutrition, and adjustable support according to the needs. In this technique, the polymer solution jet is accelerated and towed in the electric field [78]. As innovative technologies, electrospinning and 3D bioprinting make it possible to obtain multiscale, multicellular tissues and bionic structures with complex cellular structure, tissue heterogeneity, and structural and functional diversity, all with in a very complex microenvironment. Significantly, the orientation of the electrospinning fibers can provide guidance for attached cells by regulating their differentiation status and affecting their morphology, thereby promoting osteogenesis [79,80].

## 4. Shaping of Scaffold Construction

Natural materials can demonstrate exceptional mechanical characteristics and serve as valuable models for the design of microarchitectural materials. The modeling of the procedures and phenomena of bone tissue engineering is an outstanding evaluation method at the stages leading to the development of the architecture and validation of proposed in vitro experiments and in vivo models. Now that there is enough experimental data to construct plausible mathematical models of many biological control schemes, explicit hypotheses can be evaluated using computational approaches to facilitate process design. There is a methodology for biomimetic processes in vitro aimed at designing bio-artificial tissues. This methodology is based on empirical concepts of developmental biology that can be translated directly into concepts and terms of technological development [81]. According to this design methodology, the overall process consists of a series of several subprocesses, each of which repeats one of the stages of tissue development in vivo [82]. These subprocesses lead to the formation of intermediate forms of tissues, some of which exhibit modular behavior—structural stability and reliability determined by internal factors—and, therefore, can be used as the building blocks of more complex fabrics in other processes. The topological properties of networks are used to determine the degree of correspondence necessary between the in vitro processes and the corresponding in vivo processes [83]. Graph models are used to study interesting properties of several biological networks, such as connectivity, motives, modularity, or reliability [84,85]. Hybrid systems are used to control chemical processes [86,87]. For example, a mathematical model is used to describe the inflammatory and erosive processes in the affected joints of individuals suffering from rheumatoid arthritis to select a high-quality implant [88]. The modeling of physical phenomena is often based on the finite element method. Two complementary approaches are possible. This includes a homogenized macroscale model in which a solid equivalent replaces the bone structure and a high-resolution microscale model in which the minor microstructural details are included [89]. To analyze the bone structure, a homogenization procedure based on the method of an asymptotic or representative volume element is used [90]. At the same time, averaging the structure at the microscale and replacing it with a solid-state model with equivalent material properties leads to the loss of local structural information, which is essential for accurate diagnosis and treatment [91]. Simplified numerical models (finite element modeling) of the bone implant complex usually assume a perfectly bonded state, which is unrealistic in principle since the implant is never fully bonded. Contact analysis using various friction coefficients has also been used to model multiple degrees of integration in the process of osteointegration [92,93,94]. It showed some correspondence with ex vivo measurements. The construction of individual numerical models and the analysis of contact with normal contact detection and separation behavior between implants and bone allows us to make conclusions about the relationship between the coefficient of friction, bone quality, and the roughness of the implant surface [95].

Similarly, such modeling does not consider the complex nature of bone and its hierarchical structure. Adaptive manufacturing is used to consider the complex shape, porosity, and functionality of the implant. In the additive manufacturing of implants, elementary cells determine the basis for porous structures [96,97]. If we talk about craniofacial defects, they have complex anatomical shapes that are difficult to achieve intraoperatively by cutting out the bone collected from the donor site. It is necessary to apply design modeling, and the performance characteristics of reconstructed implants/prostheses can be predicted with high accuracy. The recent introduction of direct digital manufacturing technologies, which allow for the manufacture of porous implants with lattice and solid-state structures at a time based on the data of a particular patient, has opened new horizons for craniofacial surgery. The method of secondary processing in the form of molding is used to manufacture the implant itself. However, the method is complex; the process involves producing a rapid prototyping (RP) model, which requires additional costs and time [98,99]. It is possible to obtain structures with controlled and complex internal architectures and corresponding mechanical properties using RP manufacturing technologies [100]. The 3D modeling methods allow one to choose a suitable range of porous materials with respect to the density of human bones [101]. When the 3D model is ready and calculated, one can start to manufacture the product using 3D printing, thereby considering the complexity of the design and the individual characteristics of the patient [102]. Another essential characteristic of the implant is its mechanical properties. The accurate analysis of the mechanical properties could be applied to materials with uniform pore morphology, thereby providing a uniform mechanical strength.

In order to meet these specified conditions, additive manufacturing (AM) technologies are employed. AM methods make it possible to produce complex lightweight parts such as homogeneous and gradient lattice structures with variable density and porosity, which leads to different characteristics of the mechanical behavior of the layer-by-layer, compared to nongradient, lattice structures and, therefore, provides freedom of design regarding the structures [103,104,105,106]. The flexibility of AM allows for the creation of highly complex geometric structures used in biomedical implants [107,108,109,110,111]. Different structural types lead to different mechanical properties; for example, an orthogonal structure has a higher strength [112,113]. The mechanical strength of the bone structure is higher in the outer region and lower in the inner region. Therefore, a gradient porosity structure should be applied to a bionic implant. However, studies of the mechanical properties of gradient structures are still limited [114,115,116]. Gibson–Ashby models can only display the mechanical behavior of “non-gradient” cellular materials; thus, understanding and knowledge of various gradient cellular structures require further investigations [117,118].

## 5. Visualization

Today, surgeons are increasingly turning to imaging techniques. Computer-aided design (CAD) and computer-aided manufacturing (CAM) systems are being actively developed to adapt to the needs of surgeons. Such systems are specifically focused on advanced visualization tools, 3D modeling, or, better to say, virtual reality. They provide an opportunity for precise preoperative planning, thereby performing virtual osteotomy resections and designing implants for a specific patient before surgery. These virtual models can be imported into the intraoperative navigation system for the precise placement of bone segment implants [119]. Significant advances in computer vision promise faster diagnosis and treatment. With various imaging methods, for example, the results of computed tomography (CT) and magnetic resonance imaging (MRI) are recorded, and medical models of the vertebral surface are compiled to predict deformations of the material. To provide the mass personalization of orthopedic implants, a combination of several technologies is required. Computed tomography is used to create an ideal detail that is as close as possible to the original anatomy of the patient, thereby providing maximum contact with the patient’s tissue. A CT scan allows you to obtain a 3D image of the original bone structure for further 3D printing, which makes implants less painful and less likely to fail [120]. For several years, computer vision has been actively used to automatically analyze 3D medical images [121]. Real-time image scanning is used to visualize the parameters of the synthesis of medical materials. The software coverage varies depending on the experiment [122]. X-ray tomography is a widely used method for visualizing the three-dimensional architecture of bone regenerated within a porous skeletal biomaterial. This method provides quantitative volumetric analysis of the X-ray attenuating materials and tissues. At the same time, synchrotron X-ray microtomography alloys for better visualization of the structures of mineralized tissues and biomaterials with high spatial resolution. X-ray microtomography in phase contrast mode (XRPCT) makes it possible to examine soft connective tissues that are invisible to absorption contrast, while they are easily observed with phase contrast [123]. In particular, the microdiffraction method of synchrotron X-ray scanning makes it possible to obtain the structural features of bone tissue at various length scales (from atomic to nanometer) at small submicrometric sites, thereby combining the transmission of a wide pulse of scattering signals and the use of submicrometric focused beams. Thus, the possibility of the simultaneous collection of X-ray small-angle scattering (SAXS) and X-ray wide-angle scattering (WAXS) images in micrometric areas has opened up new scenarios for the X-ray examination of heterogeneous complex systems such as bone tissue and for the recognition of various components coexisting in such small areas [124]. A combination of Talbot interferometry (TI) X-ray spectroscopy and X-ray scanning microdiffraction on a single sample for the study and 3D visualization of soft tissues that contain a precursor to mineralized bone can provide important information on the first steps of biomineralization [125].

## 6. Modeling of Biodegradation

In addition to modeling implant design, an essential area in tissue engineering is modeling implant biodegradation. The design of biodegradable orthopedic implants made of biodegradable metal materials is a complex area of biomechanics [126]. Several years of experimental approaches have led to a mechanistic understanding of the degradation process. Combining this knowledge with in silico modeling approaches allows researchers to study the properties of biodegradation and implant behavior in a virtual environment before conducting any in vitro or in vivo tests. With complete validation, computer modeling can (partially) replace certain stages of expensive and long-term experiments that test implant expected destruction behavior [127]. Computational modeling of the degradation process includes various approaches, from fundamental phenomenological realizations to complex mechanistic models considering multiple aspects of the degradation and resorption process. One of the mathematical models used to estimate the sample degradation rate for a given implant geometry is based on the continuous damage (CD) theory [128]. CD theory, used in the finite element method (FE) framework, allows one to model sample degradations of various origins that are represented by different mechanisms [129]. The mechanics of continuous damage allow you to simulate the loss of mechanical strength of a material due to the presence of geometric discontinuities by defining a scalar field that quantifies the distribution of damage. Damage models are usually implemented within the framework of the FE, usually for modeling the plastic damage of metal. They can even be applied to medical devices of complex geometry. Moreover, the method proved to be an effective approach for modeling various types of corrosion [130]. The CD approach translates the presence of geometrical discontinuities into the reduction to macroscopic mechanical properties of the material (e.g., stiffness, yield stress, etc.) through the definition of a damage field by means of the continuous functions of coordinates and time [128]. An important factor is the effect of the implant on the surrounding tissues. The initial disadvantages, such as corrosion and insufficient strength of the implants, were excluded from the next versions. Additional research is needed to develop implants that accelerate fracture healing without disrupting bone physiology. For example, the introduction of denser alloys led to cortical porosis, delayed bridge formation, and refractive fractures after removal of the plate. Their unreasonable effects were caused by bone–plate contact that interfered with cerebral cortex perfusion. Consequently, further modifications of the plates were aimed at reducing this contact area to minimize necrosis and subsequent porosity. That is why it is important to understand mechanobiology to develop an orthopedic implant that improves fracture healing without interfering with bone physiology [131]. The use of biodegradable implants that do not harm the surrounding tissues can solve this problem. They are installed to support a broken bone and prevent it from shifting after the healing process. For a certain period, they perform a supporting function until the main bone is restored. Then, they are gradually dissolved or absorbed (in the form of nutrients), thereby contributing to the healing process [126,132].

## 7. Screening

There are two main categories of medical imaging: diagnostic imaging and preventive screening. The FDA defines medical imaging as any of “several different technologies that are used to view the human body for the purpose of diagnosing, monitoring, or treating diseases”. Diagnostic imaging is a group of medical imaging techniques that use noninvasive techniques to diagnose and monitor diseases. It often helps to identify the root cause of a particular symptom that is used to study a specific problem. Preventive screenings identify health problems before the symptoms of the disease develop into more serious problems. Screening is a clinical procedure that includes detection, diagnosis, and monitoring using images.

Types of screening imaging include the following:CT scans—Also called CAT scans, computed tomography (CT) scans use special X-ray equipment to take images from different angles, which are then processed by a computer to show a cross-section of body tissues and organs.Fluoroscopy—A type of imaging that shows real-time, moving X-rays of the internal body structures.Mammogram—This involves several X-ray images of the breast.MRI—Magnetic resonance imaging (MRI) uses radio waves, a strong magnetic field, and a computer to generate detailed, cross-sectional images of the patient’s internal body part.Nuclear imaging—A diagnostic tool used to accurately visualize the flow and function of different organs of the body.Ultrasound imaging (or sonography)—A method of seeing inside the human body using high-frequency sound waves.X-rays—This type of imaging uses a minimal dose of ionizing radiation to produce pictures of the body’s internal structures.

Artificial intelligence, however, can provide smart tools to screen massive amounts of medical data [133]. This significant increase highly influences the trend toward the study of DL models for medical image recognition due to the availability of medical image data. In recent years, considerable progress has been made in screening with the usage of AI in general and DL models mainly powered by convolutional neural networks (CNNs).

CNNs are DL models that are optimized for analyzing image data; they are inspired by the human visual system and employ layers with distinct roles. Convolutional layers extract intricate patterns by applying filters to the input data. The pooling layers then reduce the dimensions of the data, thereby retaining essential features. Fully connected layers integrate these features for predictions. The CNN’s hierarchical architecture and automated feature extraction makes it a perfect instrument for computer vision tasks, including image recognition and object detection.

Using plain radiographs, CNN model application can be found in automated deep learning tools for predicting bone density and fracture risk. An example of CNN application is shown on Figure 3. Such application helped to identify hip fractures, vertebral compression fractures, and morphological abnormalities while estimating the probability of a fracture occurring within the next decade. Its performance is on par with traditional methods and can potentially enhance osteoporosis screening, particularly in regions with limited access to specialized equipment. A CNN was used to extract region-of-interest (ROI) information from plain radiographs of the pelvis and lumbar spine, which was used by classification algorithms to identify specific conditions [134].

Where deep learning or other AI models with explainability issues are deemed necessary, under this governance model, interpretable frameworks are expected to enhance the decision-making process. Several medical studies have showcased how this is possible with the use of explainable tools, ranging from visual to direct measurement tools [135,136,137].

Explainable artificial intelligence (XAI) is a field that focuses on designing intelligent systems that can explain their recommendations to a human being. There are two main approaches: (a) interpretable models, which rely on non-black-box systems such as rule-based ones, and (b) prediction interpretation and justification, which aim at generating explanations for a black box algorithm. Some works mention the third one—visualization [138].

Statistical analysis shows that the software’s performance for estimating bone ages and each probability according to the attention map from an XAI model is as good in terms of the mean absolute error (MAE) and root mean squared error (RMSE) metrics as the estimation from three radiologists. Researchers verified the potential of the software based on the XAI model in clinical settings [139].

## 8. Future Directions

In order to train a neural network, large amounts of data are needed. In medical research, this is difficult for many reasons. Privacy regulations obstruct accessing and sharing medical data for research purposes. A lot of medical images are annotated and labeled manually. That is why their it is a time-consuming process to use them. The distribution of medical diseases is imbalanced, and rare conditions could be poorly represented. However, some solutions could be applied to overcome these issues. Some apparent suggestions include close collaborations with research organizations and medical institutions, conducting long-term studies to comprehend bone processes after medical treatment, and creating clear ethical regulations. In the medical field, ensuring the quality of datasets is essential for accurate and reliable outcomes. Dataset quality assurance involves thorough validation processes, including the examination of data integrity, completeness, and adherence to regulatory standards, thereby ultimately safeguarding the integrity of medical research, diagnostics, and decision-making processes.

Some new achievements could be used to obtain large datasets from small amounts of tissues.

One of the current medical achievements are the so-called organs-on-a-chip (AOAC) systems [140,141,142]. These microscale devices combine tissue engineering features with microfluidic performance. Researchers put living cells into the microfluidic device and mimic the environmental parameters affecting such cells in a body. This allows for complex simulations of tissue responses, thereby enabling implant or drug testing and studying physiological and pathological processes in the human body. While this concept was widely applied to such organs as lungs, intestines, liver, kidneys, and others [143,144,145], applying it to bone tissues is still challenging. This is due to the fact that a complex model of bone tissues includes different types of cells, such as osteocytes embedded in the mineralized organic matrix and osteoblasts located on the bone surface [146]. Moreover, the direct growth of human primary bone cells is difficult to achieve, and animal-derived cells are applied instead in some experiments. One of the recent examples is the successful growth of bone-like tissues with the shape and dimensions of human trabeculae [147]. The authors used human mesenchymal stromal cells differentiated into the osteoblastic lineage. Advances in the so-called “bone-on-a-chip” approach are summarized in recent reviews [147,148]. Despite the exciting prospects, there are not many cases in which the “bone-on-a-chip” approach was supported by artificial intelligence. To the best of our knowledge, only one article reported such a combination. Paek et al. succeeded in the growth of an osteon bone unit in a microfluidic device [146]. This bone-on-a-chip platform was further applied to test anti-SOST antibody drugs using an AI-based image analysis system. β-catenin, nucleus, and merged fluorescent images were collected from the drug-treated group and the nondrug group. The AI system exhibited an accuracy of 99.5%, thereby indicating great performance for osteoporosis drug testing.

## 9. Conclusions

The development of medical materials represents one of the most exciting new synthetic approaches to bone implant design. The introduction of artificial intelligence technologies, depicted on Figure 4, opens many opportunities for the development of a multifunctional material that meets the various needs of tissue engineering (namely, favorable chemical composition, density, adhesive surface, biological activity, etc.). To see the full picture, we suggest readers to take a look at the Table 1, which summarizes the artificial intelligence models used in bone tissue engineering (Figure 5).

Nevertheless, artificial intelligence has several concerns to be constantly addressed. Data bias is a primary concern, as it relies on data that may be skewed, thereby leading to inaccuracies, especially for underrepresented groups, different populations, or materials. Regulatory compliance, such as adhering to healthcare regulations and ethical issues related to patient data privacy, can slow down AI development and deployment. Federative learning and novel ethical regulations can be an approach to address this. Rigorous validation and testing, thereby demanding large datasets and extensive clinical trials, are essential for ensuring the accuracy of AI algorithms in bone tissue engineering.

Today, artificial intelligence accompanies all stages of the implantation process, thereby helping to minimize treatment and recovery time. The predominance of scaffolds that have been developed and those that are currently under development allow for a wide range of materials to be produced using this approach. Scaffolds can be constructed at the material level using functional concepts of polymer science but can be easily functionalized using bioactive groups at the genetic design stage. The high level of control and functionality makes porous scaffolds a compelling material platform in medicine. Methods using artificial intelligence are becoming more and more accessible to scientists. Consequently, the growing interaction between the fields of biology, chemistry, materials science, and AI inspires the invention and study of increasingly new biomaterials.

## Figures and Tables

**Figure 1 biomedicines-12-00076-f001:**
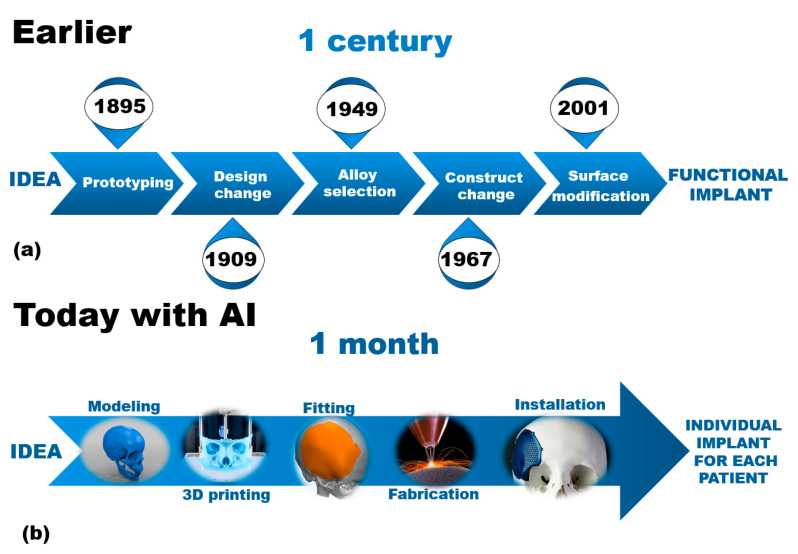
The schematic representation of workflows in medical engineering: The obsolete approach of trial-and-error with outdated approaches (**a**) vs. novel AI-assisted accelerated materials discovery (**b**).

**Figure 2 biomedicines-12-00076-f002:**
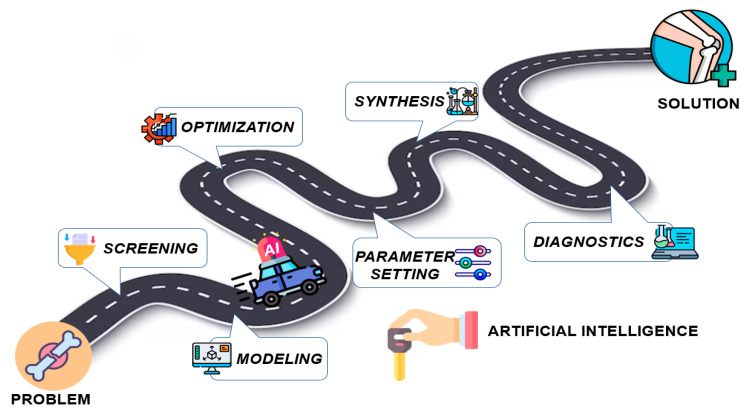
Roadmap for convergence of AI and medical chemistry for accelerated novel materials design in the field of bone tissue engineering.

**Figure 3 biomedicines-12-00076-f003:**
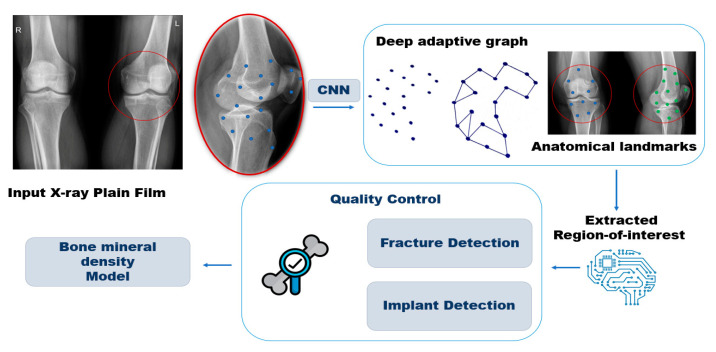
Schematic representation of the workflow for assessing bone mineral density.

**Figure 4 biomedicines-12-00076-f004:**
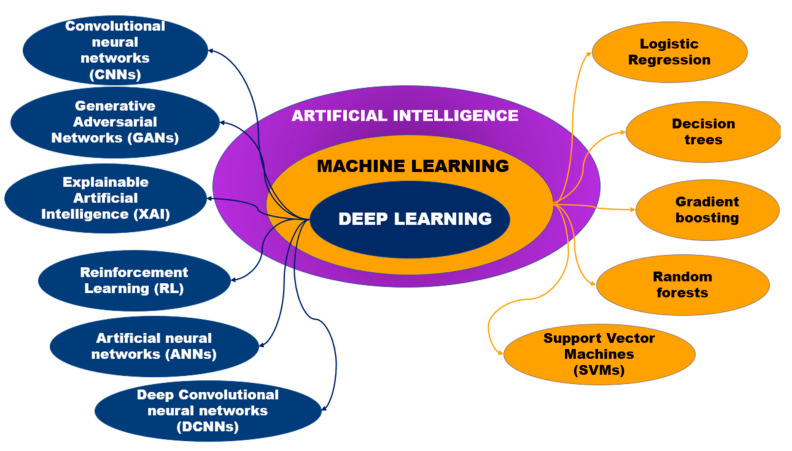
The hierarchy of AI models used in bone tissue engineering with respect to complexity.

**Figure 5 biomedicines-12-00076-f005:**
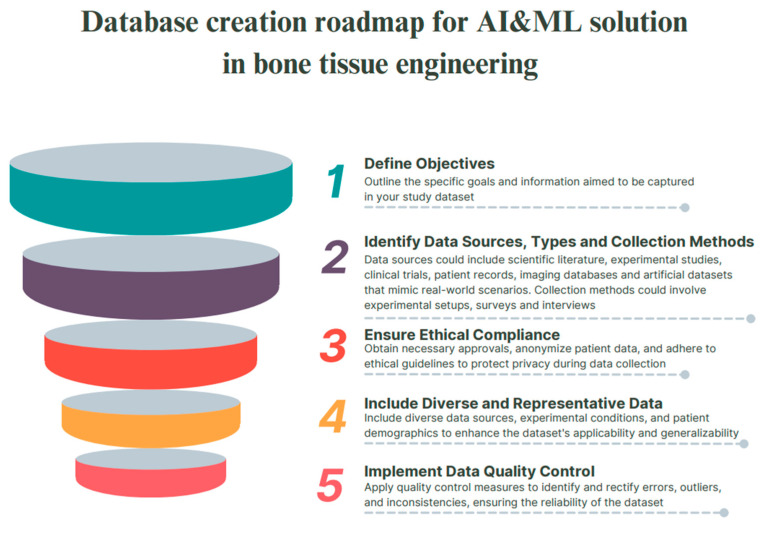
Database creation roadmap for AI and ML solutions in bone tissue engineering.

**Table 1 biomedicines-12-00076-t001:** Overview of AI models used in bone tissue engineering.

AI Model	Application	Model Key Parameters	Advantages	Disadvantages
Machine Learning Models				
Logistic Regression	Material biocompatibility, bone health assessment [149]	Weights, bias, regularization, solver	Efficiently models relationships, controls overfitting, and ensures convergence, thereby aiding in the identification of key factors	Assumes linearity, requires tuning for optimal regularization, features solver sensitivity, has potential suboptimal convergence
Support Vector Machines (SVMs)	Material identification and classification [150,151]	Kernel functions, regularization parameters	Usable for real-time assessment of scaffolding structures, particularly in high-dimensional spaces	Limited scalability to large datasets, sensitivity to the choice of kernel and hyperparameters
Decision Trees	Cell classification, Bone health sssessment [151]	Hierarchical decision structure	Easy to interpret, handles nonlinear relationships	Prone to overfitting, sensitive to small changes in data
Random Forests	Comprehensive parameter evaluation [68,149,151]	Decision trees, number of estimators, max depth	Simultaneous evaluation of multiple scaffold parameters, robust against overfitting	Prone to overfitting with noisy or imbalanced datasets, potential issues with interpretability
Gradient Boosting	Predicting biomaterial properties [151]	Learning rate, number of trees, depth of trees, loss function	Improves model accuracy by combining weak learners, captures intricate relationships, provides feature importance insights	Sensitive to noisy data, potential overfitting with limited data, computationally intensive, requires careful hyperparameter tuning
Deep Learning Models				
Artificial neural networks (ANNs)	Prediction of material properties, biocompatibility assessment of implants [69,151,152]	Nonlinear modeling, intricate parameter interactions such as cell proliferation, differentiation, material properties	Adaptability, data-driven learning, accurate prediction of complex patterns in bone tissue engineering	Large dataset requirements, potential overfitting, complex neural network structures, challenging interpretation
Convolutional Neural Networks (CNNs)	Image analysis for screening, bone mineral density prediction, and fracture risk assessment [134,153,154]	Convolutional layers, filter sizes, pooling operations	Robust feature extraction, object detection, semantic and instance segmentation	Computational intensity, reliance on large datasets, potential overfitting in certain scenarios
3D Deep Convolutional Neural Networks (DCNNs)	3D image analysis for scaffold design, screening [155,156]	Architecture, filters, voxel size	Captures intricate spatial features in 3D medical images; enhances screening accuracy	Computationally intensive for large datasets, requires substantial computational resources, potential overfitting with limited data
Generative Adversarial Networks (GANs)	Scaffold design and optimization [155,157,158]	Latent space representation, generator and discriminator architectures	Enables the synthesis of realistic scaffold structures, thereby facilitating optimization processes	Susceptible to mode collapse; potential challenges in generating clinically viable structures
Reinforcement Learning (RL)	Scaffold fabrication process optimization [159,160]	State space, action space, reward functions	Optimizes scaffold production processes through iterative learning	Limited applicability in highly dynamic or complex environments; challenges in defining reward functions
Explainable AI (XAI)	Interpretable models for bone tissue screening [135,136,137,138,139]	Local and global explanations, interpretable models, feature importance, model-specific parameters, and human-understandable representations	Provides insights into model decisions; enhances trust in screening outcomes	Trade-off between accuracy and interpretability; potential complexity in explaining deep learning models

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
