# Peer review of "Application of Artificial Intelligence at All Stages of Bone Tissue Engineering"

_biomedicines, 2023, doi:10.3390/biomedicines12010076_

Round 1

Reviewer 1 Report (Previous Reviewer 1)

Comments and Suggestions for Authors

Authors have extensively addressed the raised concerns. I will recommend for acceptance for publication.

Author Response

We are grateful for the positive evaluation of our manuscript.

Reviewer 2 Report (Previous Reviewer 2)

Comments and Suggestions for Authors

The authors have addressed the comments, and the revised manuscript has incorporated all the suggestions satisfactorily. The manuscript can be published pending editorial approval. 

Author Response

We are grateful for the positive evaluation of our manuscript.

Reviewer 3 Report (Previous Reviewer 3)

Comments and Suggestions for Authors

Title: “Application of artificial intelligence at all stages of bone tissue engineering”

In this work the authors discuss the most promising areas of AI application to the field of bone tissue engineering and prosthetics. The authors claim that it can drastically benefit from AI-assisted optimization and patient-personalization of implants and scaffolds, e.g. from visualization and real-time monitoring to the implantation cases prediction, leveraging the compromise between specific architecture decisions, material choice and synthesis procedure. The authors emphasized the crucial role of accuracy and robustness of developed AI algorithms. Indeed, the authors in bone tissues engineering claim that rigorous validation and testing, demanding large datasets and extensive clinical trials, are essential. Finally, they show how developing multidisciplinary cooperation of biology, chemistry with materials science, and AI, these challenges can be addressed.

General comment: The authors reworked their work which, now seems to be improved. However, some minor points still remain to be addressed:

1) The quality of the language should be improved. Please check the end of each line.

2) Table 1. Overview of AI models in bone tissue engineering. This table should be reworked and its size should be improved

3) Figure 4. AI techniques used in bone tissue engineering nowadays. This figures should be improved together with the caption

Comments on the Quality of English Language

The quality of the languag should be improved

Author Response

Title: “Application of artificial intelligence at all stages of bone tissue engineering”

In this work the authors discuss the most promising areas of AI application to the field of bone tissue engineering and prosthetics. The authors claim that it can drastically benefit from AI-assisted optimization and patient-personalization of implants and scaffolds, e.g. from visualization and real-time monitoring to the implantation cases prediction, leveraging the compromise between specific architecture decisions, material choice and synthesis procedure. The authors emphasized the crucial role of accuracy and robustness of developed AI algorithms. Indeed, the authors in bone tissues engineering claim that rigorous validation and testing, demanding large datasets and extensive clinical trials, are essential. Finally, they show how developing multidisciplinary cooperation of biology, chemistry with materials science, and AI, these challenges can be addressed.

General comment: The authors reworked their work which, now seems to be improved.

We are grateful for the positive evaluation of our manuscript.

However, some minor points still remain to be addressed:

1) The quality of the language should be improved. Please check the end of each line.

We worked to improve the quality of language in the manuscript.

2) Table 1. Overview of AI models in bone tissue engineering. This table should be reworked and its size should be improved

Thank you for your feedback on the table. We appreciate your suggestion, and as advised, we have made enhancements to the table by incorporating additional information for a more comprehensive overview.

3) Figure 4. AI techniques used in bone tissue engineering nowadays. This figures should be improved together with the caption

We acknowledge the need for improvement in the data presentation. Figure 4 was edited to deliver an improved and refined version.

This manuscript is a resubmission of an earlier submission. The following is a list of the peer review reports and author responses from that submission.

Round 1

Reviewer 1 Report

Comments and Suggestions for Authors

The manuscript entitled “Application of artificial intelligence at all stages of bone tissue engineering” by Kolomenskaya et al for Biomedicines Journal. This is a very nicely explained, and at very good time review article. In this review, authors have highlighted the role of artificial intelligence (AI) and help to enhance the personalized medical implants design and preparation using various techniques such as 3D printing/bioprinting, electrospinning, and molecular imprinting. Moreover, the contents of this manuscript is concise and nicely articulated. I will recommend for minor revision before acceptance for publication.

 Following concerns should be addressed thoroughly:

1.     Please revised the Introduction section with highlights of role of 3D printing/bioprinting, electrospinning for materials design and synthesis for bone tissue regeneration. Following references would be helpful: Acta Biomaterialia 2014, 10, 1238-1250, J. Mater. Chem. B 2023, 11, 6225-6248, and  Current Osteoporosis Reports 2020, 18, 505–514.

2.     Please revised the section3 “Selection of candidate’s material” with respect to 3D printing/bioprinting.

3.     Please revised the section 5 “Visualization” with bit detailed role of micro-computed tomography (µ-CT), and X-ray imaging techniques.

4.     I would be great if authors can highlight the prediction of mechanical strength of implant with time in body that would be great.

Reviewer 2 Report

Comments and Suggestions for Authors

The review article by Kolomenskaya et al describes the use of artificial intelligence in the realm of bone tissue engineering (BTE). The review is timely, and it has relevance for readers in the field of BTE, which is one of the more widely explored field in tissue engineering and regenerative medicine.

Here are few suggestions which the authors should consider for improving the manuscript:

1.     The introduction is well written however, if the authors could discuss a bit more on biomaterial selection, for instance why focus only on Hap, but rather bring the importance of selecting other appropriate bioceramics (like bioactive glass) or degradable metals or polymers for making composites, which itself is tedious if followed a conventional route. Few relevant literatures:

·     Scaffold fabrication technologies and structure/function properties in bone tissue engineering." Advanced functional materials 31.21 (2021): 2010609

·     Materials design for bone-tissue engineering." Nature Reviews Materials 5.8 (2020): 584-603.

2.     Any AI & ML strategy needs a robust database, it would be interesting if the authors could draw a roadmap on how to create a robust database from the vast literature available in literature.

3.     With the advent of third generation biomaterials which are ‘cell-instructive’ in nature, the authors can also highlight the importance of acquiring ‘Systems Biology’ approach for understanding cell-material interactions in vitro for predicting their response in vivo (in sections 4 and 7). Few interesting works as follows:

·     Developmental engineering: a new paradigm for the design and manufacturing of cell-based products. Part II. From genes to networks: tissue engineering from the viewpoint of systems biology and network science." Tissue Engineering Part B: Reviews 15.4 (2009): 395-422

·     A survey on methods for modeling and analyzing integrated biological networks." IEEE/ACM Transactions on Computational Biology and Bioinformatics 8.4 (2010): 943-958.

4.     In this regard, development of organ-on-chip models have become the norm more recently where we can predict the cell response better in response to a biomaterial cue. The authors can highlight these interesting facets in the future directions/ conclusion section.

·       "Organs-on-a-chip: A union of tissue engineering and microfabrication." Trends in Biotechnology (2023).

5.     The review can also benefit from a table listing the different numerical/ computational models (key parameters to consider in such models) that can be applied for prediction, their advantages and disadvantages for readers who are novice in the field.

Reviewer 3 Report

Comments and Suggestions for Authors

Title: “Application of artificial intelligence at all stages of bone tissue engineering”

In this work the authors discuss the most promising areas of AI application to the field of bone tissue engineering and prosthetics. The authors claim that it can drastically benefit from AI-assisted optimization and patient-personalization of implants and scaffolds, e.g. from visualization and real-time monitoring to the implantation cases prediction, leveraging the compromise between specific architecture decisions, material choice and synthesis procedure. The authors emphasized the crucial role of accuracy and robustness of developed AI algorithms. Indeed, the authors in bone tissues engineering claim that rigorous validation and testing, demanding large datasets and extensive clinical trials, are essential. Finally, they show how developing multidisciplinary cooperation of biology, chemistry with materials science, and AI, these challenges can be addressed.

General comment: The aim of this review work is interesting however, the quality of language is suboptimal and several parts of the work are not clear. Some descriptions are not clear, not detailed and not enough explaining for the interested readers. The work provide only a superficial description of the use of the AI in bone engineering. It is not clear how this work could add something to the current state of the art. Therefore, the main text of the manuscript should be rewritten to improve the quality and the impact of the work. It is not suitable for publication.

Some detailed comments:

Lines: “Among phosphorus - containing calcium salts, hydroxyapatite (Ca10(PO4)6(OH)2, 44

HAp) has the greatest similarity to the mineral part of the bone, as the thermodynamically

most stable crystalline phase of phosphorus - containing calcium salts is in body fluid [5, 46

6]. Moreover, ceramics based on HAp are biodegradable, biocompatible, and bioactive. 47

Its similarity to the inorganic part of the bone makes it an excellent alternative to auto- 48

grafts and allografts [7]. “

*) The meaning of these lines is not clear. The authors should better explain and better express what is the main content of these lines. What is meaning of :“Among phosphorus - containing calcium salts, hydroxyapatite (Ca10(PO4)6(OH)2, 44 HAp) has the greatest similarity to the mineral part of the bone, as the thermodynamically most stable crystalline phase of phosphorus - containing calcium salts is in body fluid [5, 46,6]”. Please pay attention to the exactness of the use of each chemical element or compound.

Paragraph “4. Shaping of scaffold construction”

*) This paragraph should be rewritten to improve its quality.

Lines: “However, there is no relationship between the coefficient of friction, bone quality, and the roughness of the implant surface”

*) Please explain in a more detailed way.

Paragraph: “5. Visualization”

*) Please expand and improve this paragraph

paragraph “6. Modeling of biodegradation”

*) This interesting paragraph should be improved and a more detailed description of the in silico techniques used to model the degradation should be provided: e.g the lines “ One of the mathematical models used to estimate the sample degradation rate for a given implant geometry is based on the continuous damage (CD) theory [61]. CD theory, used in the finite element method (FE) framework, allows one to model the sample degradation of various origins represented by different mechanisms [62” should detailed and the CD theory at least introduced for interested readers.

Lines: “It is needed to understand the mechanobiology to develop an orthopedic im- 243

plant that improves fracture healing without interfering with bone physiology [63]. A biodegradable implant is installed to support the broken bone and prevent its displacement 245

after the healing process. It is also gradually dissolved or absorbed (in the form of nutri- 246

ents), contributing to the healing process [59, 64]. We can conclude that artificial intelligence is deeply embedded in the field of bioengineering. It actively helps physicians improve methods of treatment and restoration of bone defects. Modeling accompanies every stage of the patient treatment: determination of implant survival, selection of suitable material, creation of an individual implant for each patient, prediction of the after-installation implant behavior, etc. “

*) Please rewrite and rework improving the meaning of not clear lines as: “It is needed to understand the mechanobiology to develop an orthopedic implant that improves fracture healing without interfering with bone physiology [63].”, etc...

Paragraph “7. Screening “

* )This paragraph should be enlarged and improved. Please rework.

Figure 4. AI techniques used in bone tissue engineering nowadays.

*) This figure is not clear. Please explain better how these AI techniques are related to the bone engineering

Comments on the Quality of English Language

The quality of the language should be improved. The abstract should be totally rewritten.